# Evaluation of Porcine Gastric Mucin-Based Method for Extraction of Noroviruses from Seaweed Salad

**DOI:** 10.3390/v17091245

**Published:** 2025-09-16

**Authors:** Philippe Raymond, Sylvianne Paul, Roxanne Blain, Neda Nasheri

**Affiliations:** 1Food Virology National Reference Centre, St. Hyacinthe Laboratory, Canadian Food Inspection Agency (CFIA), 3400 Casavant Boulevard West, St. Hyacinthe, QC J2S 8E3, Canada; 2Food Virology Laboratory, Bureau of Microbial Hazards, Food Directorate, Health Canada, 251 Sir Frederick Banting Driveway, Ottawa, ON K1A 0K9, Canada; neda.nasheri@hc-sc.gc.ca; 3Department of Biochemistry, Microbiology and Immunology, University of Ottawa, Ottawa, ON K1N 6N5, Canada

**Keywords:** human norovirus, porcine gastric mucin, magnetic beads, seaweeds, viral extraction

## Abstract

Human noroviruses (HuNov) are the major cause of foodborne illness globally. Several HuNoV outbreaks have been linked to contaminated ready-to-eat seaweed products. Standard protocols such as the ISO 15216 show limited efficiency in extracting foodborne viruses from seaweed products. Therefore, we evaluated the efficiency of an extraction protocol based on porcine gastric mucin conjugated magnetic beads (PGM-MBs) to recover HuNoVs from Wakame seaweed salad. Compared to other HuNoV extraction methods, the PGM-MB method was more efficient. We then aimed to further improve this protocol by modifying several factors such as the buffers, pH, bead concentration, centrifugation and incubation time. The optimized PGM-MB method yielded 19 ± 3% and 17 ± 4% recovery, for HuNoV GI and GII, respectively. The limit of detection (LOD_95_) for Wakame seaweed salad was 131 and 56 genomic equivalents per 25 g for HuNoV GI and GII. Although some variability in recovery efficiency was observed between the PGM sources, the optimized PGM-MB protocol effectively extracts HuNoVs from Wakame seaweed salads of various brands and other commodities such as dates, green onions, and salted seaweed. These results support the implementation of the optimized PGM-MB method as a viable alternative for HuNoV surveillance in complex food matrices.

## 1. Introduction

Human noroviruses (HuNoV) are the leading cause of foodborne outbreaks [1,2,3]. According to the Public Health Agency of Canada, 65% of known causes of foodborne illnesses in Canada are related to HuNoV [4]. In the United States, HuNoV infections account for 35% of single-etiology foodborne outbreaks and 46% of foodborne illnesses [5].

Noroviruses are small (27–40 nm), non-enveloped, single-stranded RNA viruses that belong to the *Caliciviridae* family. There are 10 distinct norovirus genetic groups [6], and most outbreaks are associated with HuNoV GI and GII. They are mainly transmitted via the fecal–oral route. These viruses can withstand a broad pH range (pH 2–9) and remain infectious for extended periods in the environment, water, on food, and surfaces (reviewed in Olaimat et al. [7], Cook et al. [8]). HuNoV can contaminate food either at the point of harvest or during food handling and processing [9].

HuNoVs are inactivated by cooking, but may remain infectious after chilling, desiccation, and drying [8,10]. While fresh and frozen produce are common vehicles for norovirus infections, HuNoVs can also be detected and remain infectious on low-moisture foods such as sun-dried tomatoes, palm dates, and dried seaweeds [11,12,13,14].

HuNoV-contaminated seaweeds have been implicated in multiple outbreaks, including an outbreak associated with seasoned green seaweeds involving 91 cases [15], and another large outbreak associated with shredded nori infecting 2094 patients [11]. Frozen Goma Wakame seaweed salads imported from China were recalled following an outbreak of HuNoV GI and GII in both Norway (100 cases) and Spain [16]. In 2024, five cases were reported in Italy, where the Ministry of Health reported the detection of HuNoV GII in Goma Wakame seaweed salad from Taiwan and issued a recall [17]. Nori is usually made from red seaweed species of the genus *Pyropia*, while Wakame is a large brown seaweed belonging to the order *Laminariales*. An FAO expert meeting on food safety concluded that HuNoVs should be considered potential microbiological hazards in seaweeds [18]. Seaweed farming in sewage-contaminated water and improper handling are likely the main routes of viral contamination [19].

The International Organization for Standardization (ISO) has established the ISO 15216 standard for the extraction and quantitative (15216–1) or qualitative (15216–2) detection of HuNoV from various soft fruits and leafy greens, stems, and bulb vegetables, based on polyethylene glycol (PEG) precipitation [20]. Since the probability of obtaining false-negative results is higher when extraction methods have low recovery rates and/or high inhibition, the ISO 15216 standard requires a minimum extraction efficiency of 1% and a maximum RT-qPCR inhibition rate of 75%. The ISO 15216 methods have not been validated for seaweed. In preliminary assays performed using the ISO 15216–1 protocol to extract HuNoV from seaweed salad (Goma Wakame), our laboratory observed gelation of the extract following the addition of guanidine thiocyanate-based lysis buffer, which precluded the elution process (unpublished). Recently, a survey was conducted to assess the suitability of the ISO 15216 standard method for HuNoV detection in seaweeds and halophytes using samples collected along the western coast of Portugal [21]. Using the Mengo virus as a process control to evaluate the extraction efficiency, these authors were able to process only half of the brown seaweed extracts. Consequently, they recommended optimizing the HuNoV extraction protocol for brown seaweed.

A potential solution to improve the selective extraction of HuNoVs at low concentrations is the use of ligands. HuNoVs bind to the fucose residues of histo-blood group antigens (HBGAs) expressed on the human gastrointestinal mucosa [22,23,24]. They also bind to porcine gastric mucin (PGM), which contains fucose residues similar to those of A and H antigens [25,26]. A high norovirus GII.4 recovery rate (47 ± 8%) from fresh green seaweed (*Enteromorpha* spp.) was reported using an approach based on PGM conjugated magnetic beads (PGM-MBs) [27]. However, the reported PGM-MB protocol needs to be adapted for the larger food portions typically used in food surveillance laboratories, as well as being tested on different varieties of seaweed-containing commodities such as seaweed salads.

Here, we report on the performance of a PGM-MB extraction protocol for Wakame seaweed. We evaluated and adapted the PGM-MB methodology reported in Suresh, Harlow and Nasheri [27] to improve its sensitivity for 25 g analytical portions, and conducted a fit-for-purpose validation for the extraction of HuNoV from Wakame seaweed salads. We also evaluated the impact of different factors on the recovery rates. Finally, we explored the potential application of this optimized PGM-MB approach for other ready-to-eat seaweed-based products and food matrices. This method should improve future food surveillance efforts and help mitigate potential food outbreaks linked to HuNoV-contaminated seaweed salads.

## 2. Materials and Methods

### 2.1. Viruses

HuNoV-positive stool samples were provided by the British Columbia Center for Disease Control. Clarified 10% stool samples were prepared as described previously [28]. Unless otherwise specified, the HuNoV GII.4 (CFIA-FVR-020, MT754281.1) and GI.5 (CFIA-FVR-022, OL345567.1) were used in the method optimization and validation. The genomic equivalent (gEq) levels of the virus aliquots were estimated following their extraction using the QIAcube and the RNeasy mini kit (Qiagen, Mississauga, ON, Canada) as described before [28]. The RNA was eluted in 50 µL RNase-free water, 1 μL of RNasin™ Plus RNase Inhibitor (Thermo Fisher, Asheville, NC, USA) was added, and the RNA was stored at −80 °C until the RT-qPCR assays were performed as described below.

In preliminary experiments, to compare the RNA extraction efficiency, the virus in 50 μL ultrapure water was heated at 100 °C for 10 min, then cooled down to 0 °C on a thermocycler and stored as described above.

### 2.2. Seaweed Samples

Samples of Wakame seaweed salads were collected from various local stores in Quebec, Canada. Unless otherwise indicated, brand A was used for method development and validation. Subsamples were prepared with 25 g ± 1 g of seaweed salad. The ingredients of the different seaweed salad brands tested are listed in Appendix A.

### 2.3. Artificial Inoculation

Unless otherwise specified, virus preparation aliquots were vortexed for 2 s and diluted in Wisent-PBS (0.005 M Na_2_HPO_4_, 0.001M KH_2_PO_4_, 0.15 M NaCl, pH 7.4, Wisent, St-Bruno, QC, Canada) at various gEq levels. One hundred µL of the diluted aliquots were spiked throughout the surface of the seaweed salad in a Whirl–Pak^®^ filter bag (VWR, Mont-Royal, QC, Canada), then left to air dry for 30 min in a biosafety cabinet. Unspiked seaweed salad subsamples were included in each extraction batch as negative controls. The amount of virus inoculated was assessed in parallel using the same RNA extraction kit as for the spiked matrices.

### 2.4. PGM-MB Preparation

Unless otherwise specified, Type III mucin from porcine stomach (catalog no. M1778, MilliporeSigma, Oakville, ON, Canada) was cross-linked to MagnaBind carboxyl derivatized beads (catalog no. 21353, Thermo Fisher) using EDC (1-ethyl-3-[3-dimethylaminopropyl] carbodiimide hydrochloride) (Thermo Fisher), according to the manufacturer’s protocol. The PGM-MBs were resuspended in 1 mL of PBS (0.05 M Na_2_HPO_4_, 0.05 M NaH_2_PO_4_, 0.15 M NaCl, pH 7.2, Fisher Scientific, Nepean, ON, Canada) with 0.05% sodium azide (MilliporeSigma) and stored at 4 °C.

### 2.5. Extraction Methodologies

#### 2.5.1. ISO 15216–1

The spiked viruses were extracted using the ISO 15216–1:2017 protocol for leafy greens and soft fruits [20]. The RNA was extracted with the NucliSens miniMAG kit (Biomérieux, Montréal, QC, Canada), following the manufacturer’s instructions. One µL of RNasin Plus RNase Inhibitor (40 U/µL) (Promega, Madison, WI, USA) was added to the eluate before storage at −80 °C.

#### 2.5.2. Magnetic Silica Beads

Viral RNAs were also extracted from Wakame seaweed salads using the magnetic silica beads (MSB) method as described before [28]. Briefly, the HuNoVs were eluted from the matrices using a Bis-Tris Propane buffer pH 8 (MilliporeSigma). The eluate was clarified by centrifugation, and pectinase was added. Magnetic silica AccuNanobeads™ (Bionneer, Oakland, CA, USA), ascorbic and malic acid (MilliporeSigma) were added. Next, HCl was added to lower the pH to 3 to bind the virus to the silica. The captured viruses were eluted from the beads by increasing the pH to 7–9. The total RNA was extracted using the RNeasy kit as described above.

#### 2.5.3. Reference PGM-MB Protocol

The protocol described by Suresh, Harlow and Nasheri [27] was used as a reference protocol for the development of the PGM-MB methodology (Figure 1). Briefly, 25 g of the inoculated food matrices were incubated with 40 mL PBS pH 7.2 and incubated at room temperature (RT) for 30 min on a rocking plate. The viral eluate was removed, and either 1 or 40 mL was incubated with 0.1 mL of PGM-MBs on a rotary platform for 30 min at RT. The 1 mL aliquots were washed using a magnet as described before [27]. The 40 mL supernatant was discarded using a 50 mL magnet rack, and the PGM-MBs were washed three times with 1 mL PBS. The total RNA was extracted from the PGM-MBs using the RNeasy kit as described above.

#### 2.5.4. PGM-MB Protocol Optimization

The impact of four different matrix elution buffers on the recovery rates was compared using PBS, TGBE buffer pH 9.5 (100 mM Tris base, Fisher scientific; 50 mM glycine, MilliporeSigma; 1% beef extract Thermo Fisher) with or without 0.05% Triton™ x-100 (Thermo Fisher), and a Tris/Glycine/pectinase buffer (100 mM Tris base; 50 mM glycine; 30 U *Aspergillus niger* pectinase, MilliporeSigma) with 0.05% Triton. Forty ml of elution buffer was added to the matrix at RT for 30 min on a rocking plate at 60 rpm. The viral eluate was removed and centrifuged for 30 min at 10,000× *g* at 5 °C. The pH was adjusted to 7.0 ± 0.5 with 5N HCl. The eluate was incubated with 0.1 or 0.2 mL of PGM-MBs on a rotatory platform at 8 rpm for 120 min at RT. Then, the eluate was incubated for 10 min on large magnetic racks, and the supernatant was discarded. The PGM-MBs were washed once with the respective elution buffer at pH 7. The total RNA was extracted from the PGM-MBs using the RNeasy kit as described above.

#### 2.5.5. The Optimized PGM-MB Protocol

The inoculated food samples were incubated with 40 mL of TGBE buffer pH 9.5 with 0.05% Triton™ x-100, 30 min at RT, on a rocking plate at 60 rpm (Figure 1). The viral eluates were removed and centrifuged for 10 min at 10,000× *g* at 5 °C. The pH was adjusted to 7.0 ± 0.5 with 5 N HCl, and the eluates were incubated with 0.2 mL of PGM-MBs on a rotatory platform at 8 rpm for 120 min at RT. The eluates were then incubated for 10 min on large magnetic racks before discarding the supernatant. The PGM-MBs were washed once with 1 mL TGBE pH 7 plus 0.05% Triton™ x-100. To elute the RNA, the PGM-MBs were suspended in 500 µL RLT buffer (Qiagen) with 2% dithiothreitol (Thermo Fisher), incubated for 2 to 5 min on the magnet rack, and extracted using the RNeasy mini kit as described above.

### 2.6. RT-qPCR Detection

HuNoV GII and GI RT-qPCR assays were carried out using the TaqMan Fast Virus 1-Step Master Mix (Thermo Fisher) as described previously [28]. Briefly, HuNoV GII RT-qPCR was performed using QNIF2d and COG2R primers and the probe QNIFS. HuNoV GI RT-qPCR was performed using QNIF4 and NV1LCR primers with the TM9 probe.

### 2.7. Recovery Rates

The recovery rates associated with the virus elution and concentration steps were estimated using the cycle threshold (Ct). The virus recovery rate = 10^(ΔCt/m)^ × 100% where ΔCt is the Ct value of extracted viral RNA from the matrix minus the Ct value of viral RNA extracted from the inoculum, and m is the slope of the virus RNA transcript standard curve [20].

### 2.8. Sensitivity

The proportion of positive observations for each concentration was used to assess the probability of detection (POD) and calculate the limit of detection (LOD_50_ and LOD_95_) and the confidence intervals (CI95%) with the PODLOD program (v9) [29]. Briefly, at each inoculum level, three to five 25 g seaweed salad subsamples were spiked with HuNoV GII.4 or GI.5 at concentrations ranging from ~10^1^ to 10^3^ gEq per 25 g. The spiked subsamples were extracted using the optimized PGM-MB protocol.

### 2.9. Inhibition Measurement

HuNoV GII RNA transcripts with insert prepared in our laboratory were used to estimate the level of inhibition as described before, by spiking a known concentration of those transcripts in the negative Wakame seaweed salad RNA extract [30].

### 2.10. Robustness

#### 2.10.1. Analyst

The impact of different analysts performing the experiments on the optimized PGM-MB recovery rates was evaluated using various Wakame seaweed salad brand A samples spiked with HuNoV GI and/or GII at 10^3^ gEq per 25 g.

#### 2.10.2. PGM-MB Preparation

Different PGM lots and brands were compared using the optimized PGM-MB method. The Biovenic Native Porcine Stomach Mucin (MUC) (Biovenic, Hauppauge, NY, USA) was compared to the MilliporeSigma Type III mucin.

In addition, the impact of PGM-MBs aging on HuNoV GI and GII recovery rates was analyzed with the Wakame seaweed salad brand A, using a PGM-MB preparation aged over a period of 16 days for GI and 48 days for GII, respectively.

#### 2.10.3. Competitor

The recovery rates of the optimized PGM-MB method when testing both HuNoV GI and HuNoV GII in competition were assessed by spiking these two viruses at various concentrations (10^3^ to 10^4^ gEq per 25 g) on the Wakame salad brand A.

#### 2.10.4. Genotype

Because the PGM-MB method requires HuNoV capsid binding, the recovery rates of a collection of different HuNoV GI (4) and GII (3) genotypes were tested in triplicate at ~10^3^ gEq with the optimized PGM-MB method.

#### 2.10.5. Seaweed Salad

Wakame seaweed salads contain multiple ingredients, which could vary between different commercial brands (Appendix A). The impact of the Wakame salad brand on the optimized PGM-MB recovery rates was evaluated using five different brands spiked with HuNoV GI and GII at ~10^3^ gEq per 25 g. Non-spiked control samples were tested in parallel.

### 2.11. Matrix Extension

A preliminary analysis of HuNoV GII.4 recovery from other Wakame seaweed types (dry Wakame, salted Wakame salad), and other food matrices (green onion, dates), was performed using 25 g of food matrices spiked at ~10^3^ gEq per 25 g.

### 2.12. Statistical Analysis

One-Way ANOVA followed by a Tukey pairwise comparison was used to compare the extraction methodology recoveries (95% CI). The Spearman correlation statistic was used to evaluate the impact of the PGM-MBs aging (*p* < 0.05).

## 3. Results

### 3.1. Reference Protocol Recovery Rates

The recovery efficiency of several published viral extraction protocols was evaluated using Wakame seaweed salad (Figure 2). The recovery efficiency was calculated as the ratio of the total viral genome copies recovered to those initially inoculated. Preliminary tests, performed either with inoculum alone or in the absence of matrices, showed that RNA recovery rates using the RNeasy kit were higher than those obtained with the boiling technique (Appendix A). Consequently, the RNeasy kit was selected for total RNA extraction from the inoculum and subsequent PGM-MB experiments. The tested HuNoV GII inoculum level ranged from 5 × 10^2^ to 6 × 10^3^ gEq. The HuNoV GII recovery rates with both the leafy green and soft fruit protocols of the ISO 15216–2017 method were low, with undetected samples and recovery rates of 1.2 ± 1.1% (2/3) and 0.1 ± 0.2% (4/9), respectively. The magnetic silica beads (MSBs) protocol, which was developed previously for leafy greens and berries, was similarly inefficient with a HuNoV GII recovery rate of 0.5 ± 0.3% (*n* = 6) [28,30]. In these preliminary assays, the reference PGM-MB protocol (1:40 mL) [27] was also applied to 25 g of seaweed salad. A 1 mL aliquot of eluate was extracted using 0.1 mL of PGM-MBs, yielding an estimated 45% recovery of the inoculated virus. Since this aliquot accounted for only 2.5% of the total eluate volume, the overall recovery rate from the inoculated matrix was 1.14 ± 0.04%.

### 3.2. PGM-MB Optimization

#### 3.2.1. Buffer Selection

We next aimed to optimize the PGM-MB method by testing whether the ISO 15216 buffer (TGBE, pH 9.5) could enhance viral extraction compared to PBS (pH 7.2), as used in the reference PGM-MB protocol [27]. Recently, Bai et al. [31] also recommended using 0.05% Triton X-100 in a Tris/Glycine/pectinase elution buffer for virus recovery from strawberries. In this set of experiments, we compared the impact of the different elution buffer compositions, with or without 0.05% Triton X-100, on HuNoV GII recovery rates, using 200 µL of PGM-MBs (Figure 3). Using PBS, the HuNoV GII recovery rate was below the 1% threshold, at 0.3 ± 0.2% (*n* = 3). Both the traditional TGBE without Triton and the Tris/Glycine/pectinase buffer with Triton yielded recovery rates above the 1% threshold at 3 ± 2% (*n* = 3) and 2 ± 1% (*n* = 3), respectively, with no significant difference between them (*p* = 0.578). In contrast, the HuNoV GII recovery rate from the seaweed salad, using the TGBE buffer with Triton, reached 8 ± 1% (*n* = 3), which is significantly higher than other buffers (*p* = 0.000). Therefore, the TGBE buffer with Triton was selected for subsequent optimization steps.

#### 3.2.2. The PGM-MB Ratio

We next evaluated the effect of the PGM-MB volume on the HuNoV recovery. When using the spiked TGBE buffer with Triton eluate from Wakame seaweed salad, the difference in recovery rates between 200 µL and 100 µL of beads (37% vs. 31%) was not statistically significant (*p* = 0.052), although a slight improvement was noted (Table 1). Therefore, 200 µL of beads was used in the remaining experiments.

#### 3.2.3. The pH Adjustment

The impact of adjusting the pH to improve the HuNoV binding to PGM-MBs was also evaluated based on the work by Tian, Brandl and Mandrell [25], who reported a significant increase in recovery at pH 3.5. For this reason, we examined whether lowering the pH to 3.5 would improve the recovery rates. As shown in Table 2, the recovery rate for HuNoV GII was significantly higher at pH 7.2 than at 3.5 (*p* = 0.0069).

#### 3.2.4. Optimization of the Centrifugation Time

We next examined whether extending centrifugation time prior to pH adjustment and beads addition would enhance the recovery rate. Using the spiked Wakame seaweed salad, the HuNoV GII recovery rates were higher at 16 ± 3% (*n* = 5) with a 10 min centrifugation at 10,000× *g* compared to 11 ± 4% (*n* = 5) with a 30 min centrifugation (*p* = 0.040) (Table 3), which demonstrates that extending centrifugation time would negatively affect viral recovery.

#### 3.2.5. Virus Incubation Time

No statistically significant difference in HuNoV GII recovery rates (17% vs. 19%) was observed when the spiking contact time and temperature (30 min at room temperature vs. 48 h at 4 °C) were compared (Table 4). On the other hand, a decrease in recovery was observed with HuNoV GI after 48 h incubation at 4 °C (*p* = 0.014). Therefore, a 30 min contact time at room temperature was selected for the remaining experiments.

#### 3.2.6. PGM-MB Aging

The effect of PGM-MB preparation aging on method performance was analyzed by comparing the recovery rates for HuNoV GI and HuNoV GII from Wakame seaweed salads over several weeks (Appendix A). HuNoV GII recovery was inversely correlated with PGM-MB aging, with an estimated half-life of 69 days (*p* = 0.000). The HuNoV GI recovery rate did not correlate with the PGM-MB preparation aging in the time frame tested (16 days, *p* = 0.364). Additional data covering a longer time frame are required for evaluating the PGM-MBs aging impact on HuNoV GI recovery.

Without considering the effect of aging on the tested PGM-MB preparations, the average recovery rates for HuNoV GI and GII from Wakame seaweed salads during the study were 15 ± 5% (*n* = 24) and 14 ± 5% (*n* = 38), respectively.

#### 3.2.7. PGM Sources

The impact of the PGM lot and type on HuNoV GII recovery rates from seaweed salad using the optimized PGM-MB protocol was explored with two different PGM type III lots from MilliporeSigma and a native PGM from a second provider Biovenic (Table 5). Upon adding EDC to the mixture of PGM and magnetic beads, the Biovenic PGM appeared more viscous than the MilliporeSigma PGM. Nevertheless, for both PGM types, the recovery rates were above 5%. However, the recovery rate using PGM-MBs with type III mucin lot A was significantly higher than with lot B or native mucin (*p* = 0.006). The type III mucin lot A was used throughout this study.

### 3.3. Validation

#### 3.3.1. LOD

The LOD_95_ values for the optimized PGM-MB extraction method from Wakame seaweed salad for HuNoV GI and GII were estimated at 131 gEq per 25 g (CI95: 56–311) and 56 gEq per 25 g (CI95: 16–193), respectively (Appendix A). The LOD_50_ values for HuNoV GI and GII were 30 gEq per 25 g (CI95: 13–72) and 13 gEq per 25 g (CI95: 4–45), respectively.

#### 3.3.2. Inhibition

Using the optimized PGM-MB negative seaweed salad control matrix (brand A) RNA extracts spiked with the HuNoV GII transcript as an external amplification control, the RT-qPCR inhibition level was estimated at 13 ± 13% (*n* = 12). The inhibition rates ranged from −1% to 40%.

#### 3.3.3. Robustness

##### Inter-Analyst Repeatability

When we compared the variation between two or three analysts in the HuNoV GI and GII recovery rates using the optimized PGM-MB method from Wakame seaweed salad, no statistically significant difference was observed (*p* = 0.09) (Table 6).

##### Competition

The optimized PGM-MB HuNoV GI and GII recovery rates from Wakame seaweed salad remained similar to the average recovery rate, whether the two viruses were spiked together or individually (Table 7). The 1 log concentration difference in the inoculum during the competition experiment had no statistically significant impact on the recovery rates (*p* = 0.089).

##### Genotype

Since the efficiency of the PGM-MB method is influenced by the capsid structure, we also evaluated its robustness with various HuNoV genotypes from genogroup I and genogroup II samples available in our collection (Table 8). In this assay, the within-genotype coefficient of variation was 27% for HuNoV GI and 70% for HuNoV GII. The average HuNoV GI recovery rate was 17 ± 5 gEq per 25 g. Except for GII.3, which was significantly lower, the average HuNoV GII recovery rate was 13 ± 5 gEq per 25 g (*p* = 0.000).

##### Seaweed Salad Brands

The recovery efficiency was evaluated using various brands of Wakame seaweed salad (Table 9). The HuNoV GI and GII recovery rates using the optimized PGM-MB method ranged from 9 to 28% with an average of 16% ± 8%. A negative control from brand D (1/3) was also positive for GII with a Ct value of 43.8. However, the amplicon could not be sequenced as the contamination level was below the method LOD. The composition of seaweed salad might also impact the recovery rate as the ingredient lists of brands B and D were nearly identical, and both showed a lower recovery rate at 9% for HuNoV GII compared to the other tested brands (Appendix A) (*p* = 0.004). There was a statistically significant difference in HuNoV GI recovery between brands (*p* = 0.000).

### 3.4. Extension

The efficiency of the optimized PGM-MB method was tested on additional high-risk matrices. The HuNoV GII recovery rates from 25 g of whole Medjool dates, green onion, and salted seaweed salad were above 10% (Figure 4). The average inhibition was 15 ± 12% for green onion. The recovery from dry Wakame was around the 1% threshold, at 1.4 ± 0.7% (*n* = 5). In these samples, 25 g of dry Wakame absorbed most of the 40 mL elution buffer, and the inhibition was relatively high at 72 ± 21%, with three out of five samples showing inhibition rates above 75%.

## 4. Discussion

Edible seaweed is available on the market in various formats, including dried or dehydrated forms (e.g., Nori sheets or flakes), semi-dried, fresh (refrigerated), preserved in water or brine, frozen, ready-to-eat products (e.g., Wakame salad), and fermented preparations. The reference PGM-MB protocol from Suresh, Harlow and Nasheri [27] was originally developed for fresh green seaweed sold in water. Dried seaweed typically represents a single ingredient matrix consumed in small portions and tends to absorb large volumes of buffer during rehydration. In contrast, the Wakame seaweed salad is a complex matrix with multiple ingredients such as oil, sugar, gum, and seeds. Additional optimization of the reference PGM-MB protocol was required. First, the portions tested were increased to 25 g, which is a common analytical portion [20]. Second, we observed that twice as much virus could be detected using the BOOM method compared to the boiling approach (Appendix A) [32]. While this might not have a major effect on the recovery rate, which is based on a ratio, it could impact the LOD estimates. Therefore, the RNeasy extraction kit was employed for RNA extraction. A third modification was to analyze the entire eluate using the PGM-MB method instead of testing 1 mL portions. Analyzing only a fraction of the food matrix reduces the method’s sensitivity and may result in an overestimation of the total virus recovery rates [33]. When using the same ratio of the PGM-MBs to a 1 mL aliquot of 40 mL eluate from the Wakame seaweed salad, we obtained a recovery rate of 45%, which was similar to the 47 ± 7.8% values reported previously [27].

The other reference protocols tested also showed limited efficiency with the Wakame seaweed salad. The PEG-based precipitation method (ISO 15216) and silica-based approaches showed limited recovery efficiency, close to or below 1%. The high concentrations of polyphenolics and polysaccharides (30–50% alginate) in brown seaweeds may have interfered with the virus recovery [34]. Moreover, the Wakame seaweed salad is a complex matrix containing multiple ingredients. The ISO 15216 reference protocol might perform better with different types of seaweed matrix. For instance, the Mengo virus RNA recovery efficiency was validated (>1%) for the majority of green and red macroalgae samples tested using the ISO 15216 protocol [21].

Several factors that could impact the recovery yields were also evaluated. Although the buffer composition was not investigated in detail, the addition of Triton significantly improved the performance of the TGBE buffer (pH 9.5) with the Wakame seaweed salad. The addition of a non-ionic surfactant may facilitate virus elution and dispersion, thus enhancing virus recovery [31]. In contrast to our findings, Bai, Pu, Suo, Zhang, Qu, Feng, Huang, Shao and Dai [31] reported a negative impact of beef extract on virus recovery from strawberries, and a 40-fold higher recovery with Tris/glycine/pectinase buffer pH 9.5 with Triton compared to TGBE with Triton (pH 9.5). Other differences, such as bead preparation, RNA extraction, or the presence of inhibitors, might explain the discrepancy in recovery compared to our study. Another important parameter is pH. The pH could affect the virus conformation and its binding to PGM-MBs. When evaluating the binding of HuNoV GII to PGM-MB without matrix, Tian et al. [35] reported a 3–4-fold increase at pH 3.6 compared to pH 7.2. In contrast, using spiked Wakame seaweed salad, in our setting, the HuNoV recovery rates increased by 1.5 to 4-fold using TGBE with Triton at pH 7.2 compared to pH 3.5. Additional differences from our study, such as the absence of matrix, buffer composition (citrate and PBS), and RNA extraction protocols, may account for the variation in recovery.

Although the different types and lots of PGM tested with the optimized PGM-MB protocol allowed an efficient HuNoV recovery (>5%), i.e., above the 1% threshold, the mucin lot variability had a noticeable impact on recovery. A decrease of approximately 3-fold in HuNoV GII recovery rate was observed between two mucin Type III lots in the same experiment. Mucin lot A, used throughout the study, consistently yielded recovery rates close to 15% with the optimized PGM-MB protocol. However, the mucin type III lot B yielded a HuNoV GII recovery rate of 5%, comparable to that of a generic mucin product lacking sialic acid specification. Accordingly, this variability should be considered as a quality control factor when PGM-MB assays are implemented.

The optimized PGM-MB protocol provided a high recovery ranging from 5% to 23%, and a low LOD_95_, close to 100 gEq, with inhibition rates below 40%. There was no significant impact of competition when both HuNoV GI and GII genogroups were co-inoculated. The assay performance was consistent across different brands and analysts. The genotype impact on the recovery rates was limited. Only one of the eight genotypes tested had a low recovery close to the 1% threshold. Genotype GII.3 is also known to be more challenging to cultivate in human intestinal enteroids [36]. A different buffer composition or additional compounds such as bile acids and ceramides might improve the mucin binding. While the ISO 15216 standard protocol does not exhibit the same genotype limitation, its performance was not superior in our assays using HuNoV GII.4. On the other hand, Hepatitis A Virus (HAV), another important human foodborne virus that is frequently surveyed with norovirus, binds to a different type of receptor than HuNoV. It uses a cellular receptor called Hepatitis A Virus Cellular Receptor 1 (HAVCR1) to enter host cells [37]. PEG-based precipitation, ultracentrifugation, silica beads, and other approaches might be more suitable for HAV extraction [38,39]. However, their performance, regarding the HAV recovery rates from the Wakame seaweed salad, remains to be evaluated.

Other PGM-MB protocols have been used previously to extract norovirus from green seaweed, lettuce, berries, and green onions [27,31,33,40]. We tested the performance of the optimized PGM-MB protocol with a limited set of samples from dates and green onions, and other seaweed products spiked with HuNoV GII. The recovery from the dates was acceptable, but 3-fold lower than the recovery yields achieved with ISO 15216 leafy green protocols [14]. In contrast, with spiked green onion, the HuNoV GII recovery rates were approximately 4-fold higher than those reported using a PEG precipitation protocol or another PGM-MB approach based on a pH 3.6 citrate binding buffer [40]. The low inhibition observed with HuNoV extracted from green onion is promising. In our hands, the ISO 15216 protocol was frequently associated with high inhibition levels when HuNoV, or the process control MNV, was extracted from green onion (data not shown).

The food portion sizes used in standard virology and microbiology assays are designed to reflect typical consumer exposure and to facilitate method comparison and result interpretation. For some low-moisture foods that require rehydration, dried matrices or spices, the consumers are certainly not exposed to the same portion sizes. The appropriate food portion size, along with the quantity and composition of the elution buffer, still needs to be determined for matrices such as dried seaweeds.

## 5. Conclusions

The optimized PGM-MB protocol significantly improves the recovery of HuNoVs from Wakame seaweed salad, achieving good recovery rates and low limits of detection. The method is robust across genotypes, brands, and analysts, though mucin lot variability should be monitored. Its potential as a valuable tool for enhanced HuNoV surveillance in other high-risk foods should also be explored.

## Figures and Tables

**Figure 1 viruses-17-01245-f001:**
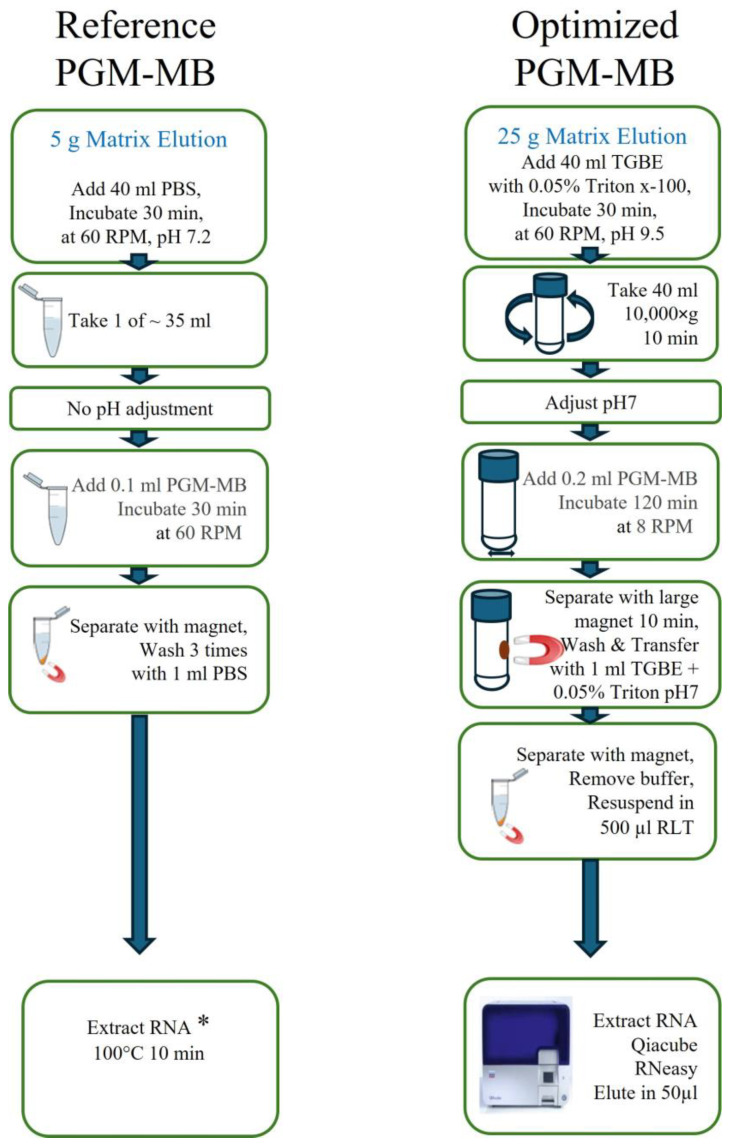
Schematic comparison of the reference protocol [27] and the optimized PGM-MB protocol used in this study. * The Qiacube RNeasy was used for RNA extraction in this study.

**Figure 2 viruses-17-01245-f002:**
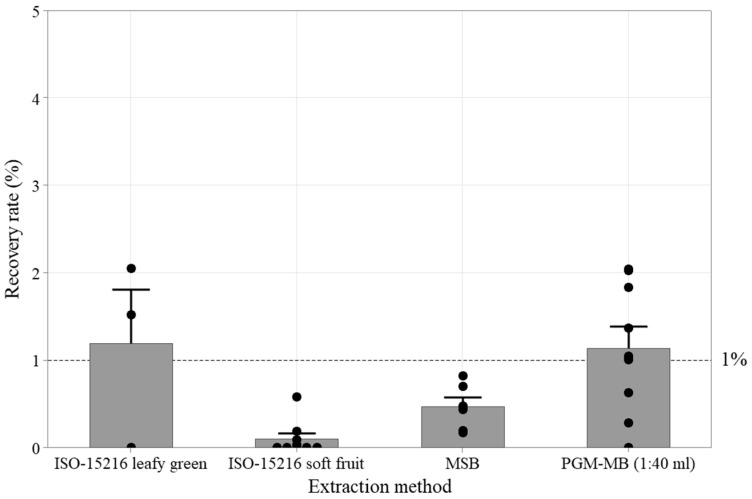
Impact of selected extraction methods on HuNoV GII recovery rates from 25 g seaweed salads. Sub-sample recovery rates (●) as well as average (bar) and standard deviation (error bar) are presented. The dotted line represents the targeted 1% recovery rate threshold.

**Figure 3 viruses-17-01245-f003:**
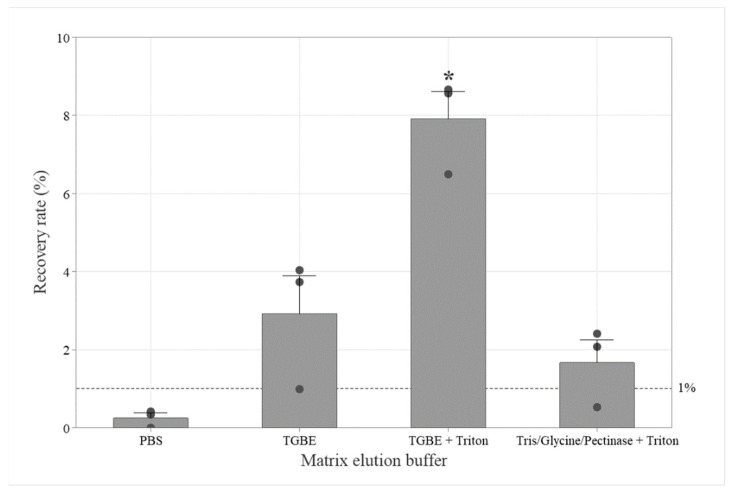
The effect of elution buffers on the HuNoV GII recovery rates from Wakame seaweed salads using PGM-MBs. Sub-sample recovery rates (●) as well as average (bar) and standard deviation (error bar) are presented. The dotted line represents the targeted 1% recovery rate threshold. The matrices were spiked with 2 × 10^3^ gEq per 25 g. The whole matrix eluates were centrifuged 10,000× *g* for 30 min and extracted with 200 µL PGM-MBs. * Indicated significant difference following Tukey post hoc comparison (*p* < 0.05).

**Figure 4 viruses-17-01245-f004:**
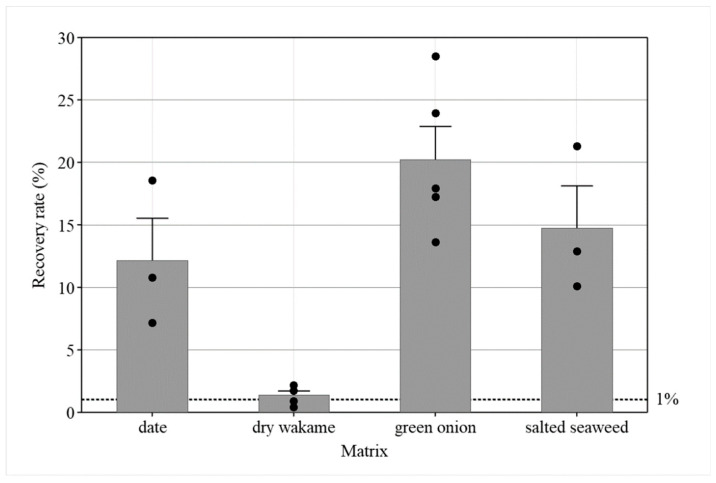
Impact of the matrix type on the optimized PGM-MB HuNoV GII recovery. Sub-sample recovery rates (●) as well as average (bar) and standard deviation (error bar) are presented. The dotted line represents the targeted 1% recovery rate threshold.

**Table 1 viruses-17-01245-t001:** The effect of PGM-MB volumes on HuNoV recovery from spiked Wakame seaweed salad extract. The Wakame seaweed salad was incubated with TGBE with Triton buffer as described in the optimized PGM-MB reference protocol. After centrifugation, the extract was spiked with HuNoV.

Beads Volume (µL)	HuNoV GII	
	Inoculum (gEq per 25 g)	Recovery Rates (Average ± sd)
100	3.1 × 10^3^	31 ± 3
200	3.1 × 10^3^	37 ± 3

*n* = 3.

**Table 2 viruses-17-01245-t002:** The effect of pH on HuNoV recovery from Wakame Seaweed using the optimized PGM-MB method.

Binding pH	HuNoV GI		HuNoV GII	
	Inoculum (gEq per 25 g)	Recovery Rates (Average ± sd)	Inoculum (gEq per 25 g)	Recovery Rates (Average ± sd)
7.2	4.2 × 10^2^	16 ± 4	4.4 × 10^3^	15 ± 3
3.5	4.2 × 10^2^	11 ± 2	4.5 × 10^3^	4 ± 2

*n* = 3.

**Table 3 viruses-17-01245-t003:** The effect of centrifugation time on HuNoV recovery from Wakame Seaweed using the optimized PGM-MB method.

Centrifugation 10,000× *g* (min)	HuNoV GII	
	Inoculum (gEq per 25 g)	Recovery Rates (Average ± sd)
10	1.6 × 10^3^	16 ± 3
30	1.6 × 10^3^	11 ± 4

*n* = 3.

**Table 4 viruses-17-01245-t004:** Impact of HuNoV GI and GII incubation time and temperature on Wakame seaweed salad recovery rates using the optimized PGM-MB method.

Incubation Time and Temperature	HuNoV GI		HuNoV GII	
	Inoculum (gEq per 25 g)	Recovery Rates (Average ± sd)	Inoculum (gEq per 25 g)	Recovery Rates (Average ± sd)
30 min @ RT	1.5 × 10^3^	19 ± 3	1.9 × 10^3^	17 ± 4
48 h @ 4 °C	1.0 × 10^3^	9 ± 2	1.5 × 10^3^	19 ± 7

*n* = 5.

**Table 5 viruses-17-01245-t005:** Impact of PGM types and lots on HuNoV GII recovery rates from Wakame seaweed salads using the optimized PGM-MB protocol.

PGM Types and Lots	Brands	Inoculum (gEq per 25 g)	Recovery Rates (Average ± sd)
Type III mucin lot A	MilliporeSigma	9.0 × 10^2^	13 ± 4
Type III mucin lot A	MilliporeSigma	9.0 × 10^2^	23 ± 3
Type III mucin lot B	MilliporeSigma	7.0 × 10^2^	5 ± 3
Native porcine stomach mucin	Biovenic	7.0 × 10^2^	5 ± 2

*n* = 4.

**Table 6 viruses-17-01245-t006:** Optimized PGM-MB HuNoV GI and GII recovery rates from Wakame seaweed salad inter-analyst repeatability.

Analyst	HuNoV GI		HuNoV GII	
	Inoculum (gEq per 25 g)	Recovery Rates Average ± sd	Inoculum (gEq per 25 g)	Recovery Rates Average ± sd
1	1.5 × 10^3^	19 ± 3 (*n* = 5)	1.91 × 10^3^	17 ± 4 (*n* = 5)
2	1.5 × 10^3^	13 ± 2 (*n* = 4)	2.3 × 10^3^	19 ± 2 (*n* = 5)
3	nt		9.0 × 10^2^	23 ± 3 (*n* = 4)

nt = not tested.

**Table 7 viruses-17-01245-t007:** The effect of competition between HuNoV GI and GII on their recovery rates from Wakame seaweed salads using the optimized PGM-MB protocol.

Inoculum Description	HuNoV GI		HuNoV GII	
	Inoculum (gEq per 25 g)	Recovery Rates Average ± sd	Inoculum (gEq per 25 g)	Recovery Rates Average ± sd
High GI/MedGII	11,183	12 ± 1	1364	12 ± 3
Med GI/High GII	1099	13 ± 5	15,624	14 ± 6
Med GI	1508	19 ± 3	na	
Med GII	na		1906	17 ± 4

*n* = 5, na = not applicable.

**Table 8 viruses-17-01245-t008:** Impact of the HuNoV genotype on the optimized PGM-MB recovery rates from Wakame seaweed salad.

Genotype	Virus Sample Number	GenBank Locus	Inoculum (gEq per 25 g)	Recovery Rates (Average ± sd)
GI.2	CFIA-FVR-001	MT745876	7 × 10^2^	13 ± 1 (*n* = 3)
GI.3	CFIA-FVR-002	MW558948	7 × 10^2^	21 ± 3 (*n* = 3)
GI.4	CFIA-FVR-003	MT750326	7 × 10^2^	12 ± 2 (*n* = 3)
GI.5	CFIA-FVR-022	OL345567	3.8 × 10^2^	19 ± 3 (*n* = 5)
GII.3	CFIA-FVR-014	MT808055	1.5 × 10^3^	1.1 ± 1 (*n* = 3)
GII.4	CFIA-FVR-018	na	1.4 × 10^3^	14 ± 3 (*n* = 3)
GII.4	CFIA-FVR-020	MT754281	1.9 × 10^3^	17 ± 4 (*n* = 5)
GII.7	CFIA-FVR-030	na	1.6 × 10^3^	7 ± 1 (*n* = 3)

na = not available

**Table 9 viruses-17-01245-t009:** Impact of Wakame seaweed salad brands on HuNoV GI and GII recovery using the optimized PGM-MB protocol.

Brands *	HuNoV GI		HuNoV GII	
	Inoculum gEq	Recovery Rates (Average ± sd)	Inoculum gEq	Recovery Rates (Average ± sd)
A	3.8 × 10^2^	19 ± 3 (*n* = 5)	1.9 × 10^3^	17 ± 4 (*n* = 5)
B	7 × 10^2^	11 ± 2 (*n* = 3)	1 × 10^3^	9 ± 2 (*n* = 3)
C	7 × 10^2^	28 ± 6 (*n* = 3)	1 × 10^3^	21 ± 7 (*n* = 3)
D **	7 × 10^2^	11 ± 2 (*n* = 3)	1 × 10^3^	9 ± 2 (*n* = 3)
E	7 × 10^2^	10 ± 2 (*n* = 3)	1 × 10^3^	26 ± 8 (*n* = 3)

* Brands A and E, or B and D, shared the same product composition. ** One spiked sample positive by RT-qPCR Ct = 43.8.

## Data Availability

The datasets generated during and/or analyzed during the current study are available from the corresponding author on reasonable request.

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
