# Peer review of "Evaluation of Porcine Gastric Mucin-Based Method for Extraction of Noroviruses from Seaweed Salad"

_viruses, 2025, doi:10.3390/v17091245_

Round 1
Reviewer 1 Report
Comments and Suggestions for Authors
Raymond et al. present a well-designed study aimed at optimizing human norovirus (HuNoV) extraction and detection from complex matrices—specifically Wakame salad—using a novel protocol.
HuNoVs are the leading cause of viral foodborne illness, and virtually any food can become contaminated at any stage of harvest, food handling or processing, posing a significant public health risk. With the increasing popularity of novel foods such as seaweed salads, there is a pressing need for sensitive and robust detection protocols capable of identifying even trace levels of viral contamination.
The first versions of the ISO 15216 standard for quantitative (15216-1) and qualitative (15216-2) detection of HuNoV were established over a decade ago for matrices including soft fruits, leafy greens, bulb vegetables, and bivalve mollusks. While widely used and extensively validated for HuNoV and hepatitis A virus (HAV), these protocols often show reduced extraction efficiency for certain complex foods. Many such foods also contain PCR inhibitors that can obscure low-level contamination.
The authors applied an optimized PGM-MB protocol to the highly challenging matrix of ready-to-eat Wakame salad. Given that seaweed consumption has been implicated in several reported HuNoV outbreaks in Asia and Europe over the past decade (with the most recent cases from Italy in 2024), and that such salads often contain complex polysaccharides, oils, and other additives that can hinder virus recovery, this work represents a particularly valuable contribution.
Through multiple optimization steps, they achieved impressive recoveries of several HuNoV genotypes (5–23%), with a low LOD95 of ~100 gEq and inhibition rates below 40%. This represents a clear improvement over the ISO 15216 standard for this matrix, in which the authors encountered gelation of the seaweed salad extract after adding the guanidine thiocyanate-based lysis buffer, rendering nucleic acid elution impossible.
The PGM-MB protocol was also tested on Medjool dates and green onions. While virus recovery from dates remained higher with ISO 15216, detection in spiked green onions was four-fold greater with the PGM-MB protocol. Given that green onions are frequently contaminated with HuNoV, this finding further underscores the method’s relevance.
I particularly appreciated the authors’ systematic approach, such as evaluating four different matrix elution buffers. I have only two minor suggestions:
- Have the authors considered re-incubating Wakame salads a second time with TGBE buffer (e.g., using 40 mL twice, combining eluates, and processing them together) to potentially boost viral recovery?
- Could PGM-MB be added directly to the TGBE buffer during initial elution? While matrix complexity and oil content might cause nonspecific bead binding or coating, testing this approach could be informative.
Overall, I recommend publication of this manuscript in Viruses. The optimized PGM-MB protocol represents a valuable advance for HuNoV detection in complex matrices, and further refinements may extend its efficiency to other challenging food products.
Author Response
Comments 1: Have the authors considered re-incubating Wakame salads a second time with TGBE buffer (e.g., using 40 mL twice, combining eluates, and processing them together) to potentially boost viral recovery?
Response 1: This is an interesting suggestion, and we have not tried this. A second incubation with TGBE buffer could increase recovery, however it could also lead to viral loss due to increase in processing volume and steps. Processing time might be a limiting factor since it could add close to 40 minutes to the full process. Nevertheless, it is certainly one parameter to investigate with dry wakame in future.
Line 517:” The appropriate food portion size, along with the quantity and composition of the elution buffer still needs to be determined for other matrices such as dried seaweeds.”
Comments 2: Could PGM-MB be added directly to the TGBE buffer during initial elution? While matrix complexity and oil content might cause nonspecific bead binding or coating, testing this approach could be informative.
Response 2: This approach was not tested with the PGM-MB method. However, without the centrifugation step, we observed clogging effects that interfered with the magnet when using magnetic silica beads for testing other matrices (28). We also expect that adding PGM-MB directly to the TGBE buffer would reduce the recovery rate due to the clogging effect and non-specific binding.
Reviewer 2 Report
Comments and Suggestions for Authors
The authors are describing evaluation of a new method for seaweed salad that builds upon detection of norovirus from other matrices, using the ISO validated methods. The authors could elaborate on the need and use of validated methods, like ISO, as many researchers and developing professionals may not understand.
Specific comments include:
In line 18 define “more” efficient.
What is reference 18?
Who is “they” in line 73? Should this be “we”?
The authors state that Wakame seaweed is the large brown kelp, but more information is useful on the salad made from this product. Is there vinegar or soy sauce? What is the pH? How is the seaweed harvested, handled, cut, prepared?
In line 128, How long were the prepared beads stored before use?
In line 165, By experience, some magnetic rack are more powerful than other, could be useful to say what product was used.
Line 277, p=0.0001?
Line 305, Stats could be added in the table by putting supscript letters
Line 322, If there is more retention of the virus when incubated at 4°C for 48h, would it be better to use those conditions when testing a recovery method?
Line 331, p=0.0001?
Table 5, What is the difference between those two lots of mucin?
Line 366, Would it be possible to have the p value for GI and the p value for GII?
Line 386, p=0.0001?
Table 8, Could the statistical difference between genotypes be added in the table using supscript letters?
Line 394, Did the sample demonstrated a log amplification at Ct 43.8?
Table 9, Could the statistical difference between genotypes be added in the table using supscript letters?
Also in Table 9, In the text, it says that it is a negative control that was positive
Author Response
Comments 1: The authors are describing evaluation of a new method for seaweed salad that builds upon detection of norovirus from other matrices, using the ISO validated methods. The authors could elaborate on the need and use of validated methods, like ISO, as many researchers and developing professionals may not understand.
Response 1: To address this comment, we added these lines to the introduction section:
Line 62. “Employing validated methods such as ISO 15216 for detection of pathogens from food commodities provides harmonized testing approaches that are recognized globally and ensures reliable and accurate results, which would support regulatory compliance and outbreak investigations.”
Comments 2: In line 18 define “more” efficient.
Response 2 : It refers to the recovery rate. As indicated line 15:” ... we evaluated the efficiency of an extraction protocol based on porcine gastric mucin conjugated magnetic beads (PGM-MB) to recover HuNoVs from Wakame seaweed salad. »
We added the information line 17.
Line 17: Compared to other HuNoV extraction methods, the PGM-MB method recovery rate was higher.
Comments 3: What is reference 18?
Response 3: FAO and WHO developed a background document identifying food safety hazards linked to the consumption of seaweed. Norovirus is one of them.
Line 588:” 18. FAO and WHO. Report of the expert meeting on food safety for seaweed – current status and future perspectives.; Rome, 28-29 October 2021, 2022.”
Comments 4: Who is “they” in line 73? Should this be “we”?
Response 4: “They” refers to “these authors” mentioned in line 76. We added reference [21] to line 77.
Comments 5: The authors state that Wakame seaweed is the large brown kelp, but more information is useful on the salad made from this product. Is there vinegar or soy sauce? What is the pH? How is the seaweed harvested, handled, cut, prepared?
Response 5: The seaweed samples are described in section 2.2 line 110. As indicated line 111 they were collected in “various local stores in Quebec, Canada”. There was no information on how they were harvested, handled, cut or prepared. As indicated line 113: “The ingredients of the different seaweed salad brands tested are listed in Supplementary Table S1.”
Comments 6: In line 128, How long were the prepared beads stored before use?
Response 6: The PGM-MBs were used between 1 to 16 days after preparation. Additional experiments were also conducted with longer storage times to evaluate the impact up to 48 days with HuNoV GII. The impact of storage time is described in section 3.2.6 “PGM-MB aging” and presented in supplementary figure S1.
Comments 7: In line 165, By experience, some magnetic rack are more powerful than other, could be useful to say what product was used.
Response 7: Two magnetic racks were used for the 50 ml tubes and one for the 1 ml tubes. The MPC-1 magnet was used for the large volumes and the Dynal magnet for the small volumes. The information was added in line 163:
Line 163: “The 40 ml supernatant was discarded using a large magnetic rack (12,200 Gauss, Dynal MPC-1, Thermo Fisher) and the PGM-MB beads were washed three times with 1 ml PBS using small magnetic rack (3,500-3,700 Gauss, DynaMag-2, Thermo Fisher).”
Comments 8: Line 277, p=0.0001?
Response 8: The statistical software Minitab reported an adjusted p=0.000, but p<0.001, is more scientifically accurate. We corrected the representation of the value lines 288, 340, 397, 410, 425.
Comments 9: Line 305, Stats could be added in the table by putting supscript letters
Response 9: When applicable (tables 2,3,4,5,8), we added an asterisk in superscript and a footnote “* Significantly lower (or higher) recovery rate (p value x)”.
Comments 10: Line 322, If there is more retention of the virus when incubated at 4°C for 48h, would it be better to use those conditions when testing a recovery method?
Response 10: A regulatory laboratory aims to rapidly assess the virus-level risk (qualitative or quantitative) of samples to which consumers have been or may be exposed, in order to help mitigate potential exposure. For certain bacteria, microbial growth can occur under specific storage conditions. Therefore, any increase during storage must be considered when evaluating consumer exposure, as outlined in testing guidelines such as the USFDA’s Guidelines for the Validation of Analytical Methods for the Detection of Microbial Pathogens in Foods and Feeds.
A lower recovery after prolonged storage may suggest the influence of other factors, such as assay variability or changes affecting virus attachment or stability. Conversely, a higher recovery after storage could indicate increased consumer exposure from the sampled matrices, compared to results obtained without storage.
However, human viruses do not grow on seaweeds or vegetables. Since no increase in viral levels was observed over time, a 30-minute room temperature storage period was selected. This information may also support method comparison efforts.
Comments 11: Line 331, p=0.0001?
Response 11: revised as suggested.
Comments 12: Table 5, What is the difference between those two lots of mucin?
Response 12: The two lots A and B come from the same provider. There are no differences expected other than the date of manufacturing and lot number.
Comments 13: Line 366, Would it be possible to have the p value for GI and the p value for GII?
Response 13: Using the ANOVA method, there were no differences between the tested viruses (p=0.085). However, if we performed the Tukey analysis of the two strains separately, there is a difference with HuNoV GI (p=0.006), and not HuNoV GII (p=0.070). However, the number of analysts that has tested the HuNoV GI is limited. We added the following comment:
Line 375: “However, the analysis of HuNoV GI recovery rate variation was based on results from only two analysts, with a difference of up to +46%. This limited dataset suggests that further investigation may be needed to better assess inter-analyst repeatability for HuNoV GI.”
Comments 14: Line 386, p=0.0001?
Response 14: Revised as suggested.
Comments 15: Table 8, Could the statistical difference between genotypes be added in the table using supscript letters?
Response 15: Revised as suggested.
Comments 16: Line 394, Did the sample demonstrated a log amplification at Ct 43.8?
Response 16: The amplification was logarithmic between Ct 30 and 37 then flattened out. Below 37 Ct the log amplification curves were parallel between controls and samples. The amplification threshold was 0.067. At 1 copy per PCR, the impact on the spiked matrix recovery was approximately less than 3%.
Comments 17: Table 9, Could the statistical difference between genotypes be added in the table using supscript letters?
Response 17: The results of the ANOVA test are presented in Table 9 which indicate the impact of the brands on recovery rates. In the revised manuscript, the statistical analysis was performed separately for each genotype using the Tukey pairwise comparison and the results are added to the table along with a footnote on line 415.
Comments 18: Also in Table 9, In the text, it says that it is a negative control that was positive
Response 18: Indeed, one of the three non-spiked samples had a very low positive signal at 43.8 Ct, which could indicate natural contamination, while the spiked positive controls had a positive signal at 32 Ct (~40 copies). The impact on recovery rate estimates was negligible, as it was less than 3%.
Reviewer 3 Report
Comments and Suggestions for Authors
This study focuses on investigating the pretreatment technology for HuNoV detection. In the HuNoV detection process, affected by the diversity and complexity of samples, pretreatment technology has always been a key bottleneck restricting detection efficiency and accuracy. Although the technology of porcine gastric mucin-conjugated magnetic beads (PGM-MB) is relatively mature, its application in extracting noroviruses from complex matrices such as Wakame seaweed salad still holds significant practical significance and research value. The core innovation of this technology lies in the first expansion of the PGM-MB method to the detection scenario of large sample size (25 g). Through multi-parameter optimization, the authors significantly improved the recovery rates of HuNoV (GI and GII), and systematically verified the robustness and multi-matrix applicability of this method. The overall research design is complete, and the data provide relatively sufficient support for the conclusions. However, there are several issues that require the authors' consideration:
- In the section "2.4 PGM-MB Preparation", the quality of PGM-MB directly affects the virus recovery rate (e.g., the recovery rate difference of PGM from different sources reached 3-fold in subsequent experiments). It is recommended that the authors describe the key preparation steps in detail to ensure the reproducibility of the experiment.
- Line 150, the expression "1 or 40 ml" is suspected to be incorrect.
- In different optimization experiments (such as buffer screening, pH adjustment, and centrifugation time optimization), there are differences in the inoculated viral genomic equivalents (gEq) (e.g., ranging from 10² to 10⁴ gEq per 25 g). The authors need to explain the basis for this design to rule out the possibility that "differences in copy numbers interfere with the optimization results".
- In the optimization of "3.2.2 The PGM-MB ratio", the virus recovery rate reached 31%-37%, which is higher than that in other optimization links (e.g., the maximum recovery rate of 8% in buffer screening and 16% in pH adjustment). The authors need to explain the reason for this difference.
- Regarding the stability of the PGM-MB method across different laboratories, it is recommended that the authors supplement some safeguard measures.
Author Response
Comments 1: In the section "2.4 PGM-MB Preparation", the quality of PGM-MB directly affects the virus recovery rate (e.g., the recovery rate difference of PGM from different sources reached 3-fold in subsequent experiments). It is recommended that the authors describe the key preparation steps in detail to ensure the reproducibility of the experiment.
Response 1: In the revised version, the section "2.4 PGM-MB Preparation" was modified to include the key manufacturer steps on line 125:
Line 125: “Unless otherwise specified, Type III mucin from porcine stomach (PGM, catalog no. M1778, MilliporeSigma, Oakville, ON, Canada) was cross-linked to MagnaBind carboxyl-derivatized beads (catalog no. 21353, Thermo Fisher) using EDC (1-ethyl-3-[3-dimethylaminopropyl] carbodiimide hydrochloride) (Thermo Fisher), according to the manufacturer’s protocol. Briefly, 1 mL of MagnaBind beads was washed three times with 1 mL of PBS (0.05 M Na2HPO4, 0.05 M NaH2PO4, 0.15 M NaCl, pH 7.2, Fisher Scientific, Nepean, ON, Canada), with gentle agitation after each wash and magnetic separation. The PGM solution (10 mg/mL) was prepared in conjugation buffer (0.1M MES, catalog no.28390, Thermo Fisher) and added to the washed beads. Separately, 10 mg of EDC was dissolved in 1 mL of conjugation buffer immediately before use, and 0.1 mL of this solution was added to the bead-protein mixture. The reaction was gently agitated and incubated for 30 minutes at room temperature. After incubation, the beads were magnetically separated, the supernatant aspirated, and the coupled beads were washed three times with 1 mL of PBS. The PGM-MB was resuspended in 1 mL of PBS with 0.05% sodium azide (MilliporeSigma) and stored at 4 °C.”
Comments 2: Line 150, the expression "1 or 40 ml" is suspected to be incorrect.
Response 2: The expression refers to the reference assay which tested 1 ml aliquot or the whole 40 ml eluate. For clarity, we modified the sentence the following:
Line 160:” The viral eluate was removed, and either the entire eluate or a 1 ml aliquot was incubated with 0.1 ml of PGM-MB on a rotary platform for 30 min at RT.”
Comments 3: In different optimization experiments (such as buffer screening, pH adjustment, and centrifugation time optimization), there are differences in the inoculated viral genomic equivalents (gEq) (e.g., ranging from 10² to 10⁴ gEq per 25 g). The authors need to explain the basis for this design to rule out the possibility that "differences in copy numbers interfere with the optimization results".
Response 3: Very low spiking level could generate more variable results than higher levels, due to variability in both the inoculum and recovery estimates. In addition to the virus extraction from the matrix, there is an inherent variability associated with the inoculum preparation steps such as vortexing, diluting the viral stock before each experiment, and its quantification. Nevertheless, since we expected that contaminated food would contain very low level of virus, we designed the experiment that allows viral detection at low concentrations. Since only 1/5 of the RNA extract is tested by RT-qPCR, we targeted a spiking level of 2x103 with 5% recovery to assure detection. Some assays, such as those for the LOD determination, were designed to test the lowest concentrations, while others, like the competition assays, targeted higher levels. Reference methodologies with recovery below 1% required higher virus spiking. Excluding these assays, the average inoculum for HuNoV GI and GII in the study was 102.9 (102.6 -103.2) and 103.2 (103.0 -103.7), respectively, which is close to the original plan.
In our study we did not observe an impact of concentration. As indicated in line 385: “The 1 log concentration difference in the inoculum during the competition experiment had no statistically significant impact on the recovery rates (p=0.089).“ We are also confident in the robustness of the approach based on inter-analyst reproducibility, and the impact of brand and genotype. Furthermore, there was no impact of the inoculum concentration tested in those assays (p>0.9) in the range tested for HuNoV GI (102,6 to 103.2 gEq) and HuNoV GII (103-103.4gEq).
Regarding the optimization phase, the concentration tested during optimization of beads volume, pH, centrifugation time and incubation time parameter was relatively stable within each parameter comparison (CV<16%). However, we cannot exclude the possibility that using higher concentrations could yield different results, as further optimization might increase the recovery rates.
We added the following sentences to explain the inoculum level design, impact of spiking level difference between study, and potential for further improvement on lines 119, 458, 473, respectively:
Line119: “Unless otherwise specified, virus preparation aliquots were vortexed for 2 s and diluted in Wisent-PBS (0.005 M Na2HPO4, 0.001M KH2PO4, 0.15 M NaCl, pH 7.4, Wisent, St-Bruno, QC, Canada) at various gEq levels around 2 × 103 gEq per 25 g.”
Line 458: “Several factors that could impact recovery yields were also evaluated, and there is potential for further improvement.”
Line 473: “Additional differences from our study, such as the absence of matrix, spiking levels, buffer composition (citrate and PBS), and RNA extraction protocols, may account for the variation in recovery. “
Comments 4: In the optimization of "3.2.2 The PGM-MB ratio", the virus recovery rate reached 31%-37%, which is higher than that in other optimization links (e.g., the maximum recovery rate of 8% in buffer screening and 16% in pH adjustment). The authors need to explain the reason for this difference.
Response 4: In this case, as described in the result (line 297) and table legend (line 303), the extract was spiked with HuNoV after centrifugation, not the matrix itself. This could explain the difference compared to other results from spiked matrix experiments.
Line 297: “When using the spiked TGBE buffer with Triton eluate from Wakame seaweed salad...”
Line 303: “The effect of PGM-MB volumes on HuNoV recovery from spiked Wakame seaweed salad extract. The Wakame seaweed salad was incubated with TGBE with Triton buffer as described in the optimized PGM-MB reference protocol. After centrifugation the extract was spiked with HuNoV.”
Comments 5: Regarding the stability of the PGM-MB method across different laboratories, it is recommended that the authors supplement some safeguard measures.
Response 5: To address this comment, we added the following cautionary statement on line 489:
Line 489: “The potential variation in the performance of the assay between laboratories was not investigated in this study.”